# Cardiac Functional and Structural Abnormalities in a Mouse Model of CDKL5 Deficiency Disorder

**DOI:** 10.3390/ijms24065552

**Published:** 2023-03-14

**Authors:** Manuela Loi, Stefano Bastianini, Giulia Candini, Nicola Rizzardi, Giorgio Medici, Valentina Papa, Laura Gennaccaro, Nicola Mottolese, Marianna Tassinari, Beatrice Uguagliati, Chiara Berteotti, Viviana Lo Martire, Giovanna Zoccoli, Giovanna Cenacchi, Stefania Trazzi, Christian Bergamini, Elisabetta Ciani

**Affiliations:** 1Department of Biomedical and Neuromotor Sciences, University of Bologna, 40126 Bologna, Italy; 2Department of Pharmacy and Biotechnology, University of Bologna, 40126 Bologna, Italy

**Keywords:** CDKL5 deficiency disorder, mouse model, prolonged QTc interval, heart aging, mitochondrial dysfunction

## Abstract

CDKL5 (cyclin-dependent kinase-like 5) deficiency disorder (CDD) is a severe neurodevelopmental disease that mostly affects girls, who are heterozygous for mutations in the X-linked *CDKL5* gene. Mutations in the *CDKL5* gene lead to a lack of CDKL5 protein expression or function and cause numerous clinical features, including early-onset seizures, marked hypotonia, autistic features, gastrointestinal problems, and severe neurodevelopmental impairment. Mouse models of CDD recapitulate several aspects of CDD symptomology, including cognitive impairments, motor deficits, and autistic-like features, and have been useful to dissect the role of CDKL5 in brain development and function. However, our current knowledge of the function of CDKL5 in other organs/tissues besides the brain is still quite limited, reducing the possibility of broad-spectrum interventions. Here, for the first time, we report the presence of cardiac function/structure alterations in heterozygous *Cdkl5* +/− female mice. We found a prolonged QT interval (corrected for the heart rate, QTc) and increased heart rate in *Cdkl5* +/− mice. These changes correlate with a marked decrease in parasympathetic activity to the heart and in the expression of the *Scn5a* and *Hcn4* voltage-gated channels. Interestingly, *Cdkl5* +/− hearts showed increased fibrosis, altered gap junction organization and connexin-43 expression, mitochondrial dysfunction, and increased ROS production. Together, these findings not only contribute to our understanding of the role of CDKL5 in heart structure/function but also document a novel preclinical phenotype for future therapeutic investigation.

## 1. Introduction

CDKL5 (cyclin-dependent kinase-like 5) deficiency disorder (CDD; OMIM 300203) is a rare and severe X-linked neurodevelopmental disease caused by mutations in the *CDKL5* gene, which lead to a lack of CDKL5 protein expression or function [1,2,3]. CDD represents one of the most common genetic causes of epilepsy in infants [4], affecting females, who are heterozygous for CDKL5 deficiency due to random X-chromosome inactivation four times more often than males [5]. Due to the numerous clinical features that overlap with the more well-characterized Rett syndrome (RTT), it was initially termed as an “early seizure variant” or “Hanefeld variant” of RTT [6]. Similarities between RTT and CDD include severe neurodevelopmental and motor impairment, intellectual disability, and, in some cases, respiratory dysregulation. Nevertheless, the improvement of the clinical overview of CDD in the past few years has defined a more detailed phenotypic spectrum, including very common alterations of peripheral organ and tissue function, such as gastrointestinal problems, irregular breathing, hypotonia, respiratory dysrhythmias, and scoliosis—complications that strongly impair the quality of life of patients and their families [7].

Several knockout (KO) mouse models for *Cdkl5* have been developed to address how CDKL5 dysfunction leads to neurological defects in CDD [8,9,10,11,12,13]. In particular, *Cdkl5* KO mouse models exhibit several behavioral phenotypes that mimic CDD features, such as impaired learning and memory, social interaction, and motor coordination, together with increased stereotypy. Cortical visual response is severely impaired in *Cdkl5* KO mice [14], supporting the idea that CDKL5 is needed for the proper functioning of cortical circuits. In this line, dendritic hypotrophy of cortical and hippocampal neurons and abnormal maturation and density of dendritic spines have been identified in *Cdkl5* KO mice [8,15,16], and these influence excitatory synaptic transmission and plasticity [16,17]. Interestingly, *Cdkl5* KO mice are characterized by accelerated neuronal senescence/death during aging [18] and by a generalized status of microglial overactivation in the brain that worsens during aging [19]. The finding that microglial overactivation exerts a harmful action in the *Cdkl5* KO brain [19,20], impairing brain development and neuronal survival, suggests that a neuroinflammatory process contributes to the pathogenesis of CDD.

CDD symptomatology [5,21], along with high CDKL5 expression levels in the brain [22,23], underscores the critical role that CDKL5 plays in proper brain development and function. However, the role of CDKL5 in biological processes that take place in non-neuronal tissues has recently emerged. Interestingly, a recent study using next-generation sequencing identified CDKL5 as a gene that is potentially involved in cardiac disorders associated with epilepsy [24], suggesting that CDKL5 may play a role in cardiac function. Moreover, two recent studies reported arrhythmia and prolonged QTc interval in some CDD patients who underwent ECG [25,26], suggesting the presence of cardiac abnormalities in the CDKL5 deficiency condition. Similarly, it has been well-established that RTT patients are more prone to display a prolonged QTc interval [27], possibly due to immature medullary cardio inhibition and reduced cardiac vagal tone [28].

Based on these premises, the overall goal of this study was to carry out the first in-depth investigation into the effect of CDKL5 loss at a cardiac level by exploiting a mouse model of the pathology. Our results demonstrate that heterozygous Cdkl5 deficiency affects cardiac function and cardiac gene expression. We also show that the absence of Cdkl5 has an important implication for mitochondrial integrity/function, promoting oxidative stress and cardiac fibrosis in the adult heart.

## 2. Results

### 2.1. Prolonged QTc Interval and Elevated Heart Rate in Cdkl5 +/− Mice

We found that *Cdkl5* is expressed in the mouse heart (Figure 1A), albeit at lower levels than in the brain. *Cdkl5* expression was reduced to 39 ± 5.6% in heterozygous *(Cdkl5* +/−) female mice in comparison with wild-type (*Cdkl5* +/+) mice (Figure 1B) and was absent in *Cdkl5* −/− mice (Figure 1B).

To determine whether CDKL5 has a physiological function in the heart, we recorded and analyzed the ECG signal during sleep in *Cdkl5* +/−, *Cdkl5* −/−, and wild-type (*Cdkl5* +/+) mice. Interestingly, we found that both *Cdkl5* +/− and *Cdkl5* −/− female mice exhibited tachycardia (reduced RR interval; Figure 1C and Table 1) and a longer QTc interval (Figure 1D and Table 1) in comparison with *Cdkl5* +/+ mice.

As expected, tachycardia also produced significant reductions in P wave duration and in QRS and JT intervals in *Cdkl5* +/− mice compared to *Cdkl5* +/+ mice (Appendix A). Using validated mouse indices computed in the time domain on spontaneous RR fluctuations (pNN8 and MRSSD), we explored the vagal (parasympathetic) contribution to cardiac modulation (Figure 1E,F and Table 1). Both indices were significantly reduced in *Cdkl5* +/− and *Cdkl5* −/− mice compared to *Cdkl5* +/+ mice. No significant difference was found between *Cdkl5* +/− and *Cdkl5* −/− mice for any of these parameters. Since CDD female patients are all heterozygous for CDKL5 deficiency, subsequent analyses were carried out in heterozygous *Cdkl5* +/− female mice. In order to investigate the influences of the autonomic nervous system on heart activity and to dissect the relative contribution of each autonomic arm in *Cdkl5* +/− mice, we evaluated heart responses to selective pharmacological blockades of the parasympathetic and sympathetic systems elicited by IP injection of atropine or atenolol. As expected, the constant IP infusion of atenolol in *Cdkl5* +/+ mice produced a significant increase in RR intervals, while atropine induced a significant decrease compared to saline infusion (Figure 1G and Appendix A). Accordingly, in *Cdkl5* +/+ mice, these drugs produced a significant increase (atropine) or decrease (atenolol) in QTc intervals compared to saline (Figure 1H and Appendix A). Interestingly, while an effect on RR and QTc intervals was elicited in *Cdkl5* +/− mice by atenolol infusion (Figure 1G,H), atropine did not produce any significant variation of RR or QTc intervals compared to saline in *Cdkl5* +/− mice (Figure 1G,H).

To investigate whether the rate variability in *Cdkl5* +/− mice is associated with an altered vagal/sympathetic component, we analyzed the expression of muscarinic 2 (*Chrm2*) and adrenergic beta 1 (*Adrb1*) receptors. We found that the mRNA levels of *Chrm2* were significantly lower in the hearts of *Cdkl5* +/− female mice than in those of wild-type (*Cdkl5* +/+) mice (Figure 1I), while no difference in the expression of the *Adrb1* receptor was observed between *Cdkl5* +/+ and *Cdkl5* +/− female hearts (Figure 1J).

### 2.2. Alterations of Voltage-Gated Channel Expression in the Hearts of Cdkl5 +/− Mice

To investigate the molecular alterations underlying the prolonged QTc interval and elevated heart rate in *Cdkl5* +/− mice, we analyzed the expression of different voltage-gated channels involved in the action potential of cardiac myocytes. In particular, we focused our attention on the genes encoding for the voltage-gated potassium (*Kcnq1*, *Kcnh2*) and sodium (*Scn5a*) channels. Mutations in these genes are the most common causes of inherited long QT interval syndrome (LQTS), a condition characterized by a prolongation of the QT interval on the ECG, an increase in heart rate, and the risk of ventricular arrhythmias [29].

RT-qPCR analyses revealed no differences in the expression of *Kcnq1*, *Kcnh2*, or *Kcnj2* channels between *Cdkl5* +/+ and *Cdkl5* +/− female hearts (Table 2); however, we did find that the mRNA levels of *Scn5a* were significantly lower in the hearts of *Cdkl5* +/− female mice than in those of wild-type (*Cdkl5* +/+) mice (Table 2).

To rule out any potential abnormalities in the sinoatrial node as possible contributing factors to the observed sinus tachycardia, *Hcn4* expression was examined. Interestingly, we found that *Hcn4* mRNA levels were significantly lower in *Cdkl5* +/− female mice than in wild-type mice (Table 2).

### 2.3. Increased Cardiac Fibrosis in Cdkl5 +/− Mice

To determine whether the absence of *Cdkl5* could impact the heart morphological structure in *Cdkl5* KO mice, dissected hearts were weighed. Since the weight of the heart positively correlates with body weight and growth, each animal’s body weight was measured before sacrifice, and the ration of heart weight to body weight (HW/BW) was calculated. No significant differences in heart weight, body weight, or HW/BW were found between *Cdkl5* +/− and *Cdkl5* +/+ mice (Table 3). Similarly, atrioventricular distance (Table 3) and the gross morphology of hearts (data not shown) revealed no differences in the chamber diameters of wild-type (*Cdkl5* +/+) and *Cdkl5* +/− mouse hearts.

Heart sections stained with hematoxylin and eosin showed normal architecture of cardiac myocytes (Figure 2A). However, Masson’s trichrome-stained sections showed increased collagen deposition in the hearts of *Cdkl5* +/− mice compared with wild-type (*Cdkl5* +/+) mice (Figure 2B). Increased fibrosis was accompanied by increased expression of fibrotic genes such as the α-2 chain of collagen type 1 (*Col1a2*) and the α-3 chain of collagen type 2 (*Col3a1*) (Figure 2C).

Given that cardiac fibrosis is mainly the result of proliferation and trans-differentiation of cardiac fibroblasts, we measured the content of vimentin, a marker of cardiac fibroblasts [30]. Western blot analysis showed significantly higher vimentin levels in the hearts of *Cdkl5* +/− mice in comparison with wild-type (*Cdkl5* +/+) mice (Figure 2D).

### 2.4. Altered Gap Junction Organization and Connexin Expression in the Hearts of Cdkl5 +/− Mice

The electrical conduction in cardiac muscle relies on efficient gap-junction-mediated intercellular communication between cardiomyocytes, which involves rapid anisotropic impulse propagation through connexin (Cx)-containing channels, namely Cx43, the most abundant Cx in the heart.

Western blotting analysis revealed significantly diminished expression of Cx43 in *Cdkl5* +/− mice in comparison with *Cdkl5* +/+ mice (Figure 3A,B). The mean density of the Cx43 band in Western blotting was reduced by 24% in *Cdkl5* +/− mice compared to the *Cdkl5* +/+ group (Figure 3B). In parallel, the mRNA levels of Cx43 did not differ between *Cdkl5* +/− and *Cdkl5* +/+ mice (Figure 3C), suggesting a post-transcriptional mechanism of Cx43 downregulation. Immunofluorescence labelling showed Cx43 expression to be mainly located at the intercalated discs (Figure 3D). As compared to *Cdkl5* +/+ mice, the Cx43 fluorescence in *Cdkl5* +/− mice was slightly less intensive (Figure 3D). It is interesting to note that some degree of lateralization of the Cx43 expression on the cardiomyocytes of *Cdkl5* +/− mice was also present (Figure 3D).

Interestingly, the expression of β-catenin protein, a component of the adherens junction of the intercalated disc, was increased in the myocardium of *Cdkl5* +/− mice (Figure 3E–H). Western blot analysis revealed that β-catenin levels were increased by 71.9% in *Cdkl5* +/− mice in comparison with *Cdkl5* +/+ mice (Figure 3E,F), which correlated with the much stronger immunoreactivity for β-catenin at the intercalated discs in the hearts of *Cdkl5* +/− mice (Figure 3G,H).

### 2.5. Increased AKT Activation in the Hearts of Cdkl5 +/− Mice

Recent evidence has described a potential role of AKT in aging-associated organ deterioration including cardiac hypertrophy and fibrosis [31]. In particular, aging enhances AKT phosphorylation in the mouse heart, while GSK-3β phosphorylation levels are unaffected by aging or AKT overexpression [31]. Evaluation of the phosphorylation levels of the AKT/GSK-3β pathway showed higher phosphorylation levels of AKT in the hearts of *Cdkl5* +/− mice in comparison with *Cdkl5* +/− mice (Figure 4A,B) but no difference in phosphorylated GSK-3β levels between *Cdkl5* +/+ and *Cdkl5* +/− mice (Figure 4C,D). Similarly, there was no difference in phosphorylated extracellular regulated kinases (ERK) 1 and 2 in the hearts of *Cdkl5* +/− mice in comparison with *Cdkl5* +/+ mice (Figure 4C,D).

### 2.6. Mitochondrial Dysfunction in the Hearts of Cdkl5 +/− Mice

There is growing evidence that mitochondrial dysfunction contributes to the development and progression of cardiac fibrosis [32]. To address whether mitochondrial function is affected in the Cdkl5-deficient heart, we first measured the ATP and ADP content in *Cdkl5* +/− mouse heart homogenates using HPLC. This analysis detected a 35% decline in total ATP content in heart tissue from *Cdkl5* +/− mice compared to controls (Figure 5A).

Moreover, in *Cdkl5* +/− mice, we found a significant decrease in the ATP/ADP ratio compared to wild-type animals (Figure 5B), suggesting a diminished cell energy status.

Since mitochondria are the major site of ATP production within the cell, we evaluated whether there were alterations in the process of oxidative phosphorylation (OXPHOS) in *Cdkl5* +/− mice. For this purpose, we isolated intact mitochondria from mouse hearts and evaluated the oxygen consumption in the presence of glutamate–malate or succinate under non-phosphorylating conditions (state 4) and in the presence of ADP (state 3, phosphorylating condition). Glutamate–malate and succinate donate electrons to the respiratory chain at complex I and complex II, respectively. State 3 of respiration, in the presence of glutamate-malate or succinate, was significantly decreased in cardiac mitochondria from *Cdkl5* +/− compared to wild-type mice (Figure 5C). In accordance with the oxygen consumption data, the spectrophotometric analysis of the single respiratory enzyme activity in isolated cardiac mitochondria provided evidence supporting reduced activity of succinate CoQ reductase (Complex II) in *Cdkl5* +/− mice in comparison with *Cdkl5* +/+ mice (Figure 5D). The complex I activity in *Cdkl5* +/− heart mitochondria showed a trend of reduction compared to controls, although it did not reach statistical significance (Figure 5D).

The decrease in the activity of mitochondrial respiratory chain enzymes prompted us to assess whether overall mitochondrial content, size, and distribution were altered in the hearts of *Cdkl5* +/− mice. We found no significant difference in the density of mitochondria in the myocytes (data not shown), nor was there any difference in the overall size of individual mitochondria (Figure 5E). However, crystalline-like inclusions in the intracristal space were found in mitochondria of *Cdkl5* +/− mice (Figure 5E), indicating mitochondrial morphological abnormalities in the *Cdkl5* +/− condition.

### 2.7. Increased ROS Production in the Hearts of Cdkl5 +/− Mice

Mitochondrial dysfunction is often associated with increased oxidative stress [33]. The level of reactive oxygen species (ROS)production determined using the fluorogenic probe DFCDA was significantly higher in *Cdkl5* +/− mitochondria energized with succinate in comparison with the *Cdkl5* +/+ condition (Figure 6A). Moreover, ROS production in *Cdkl5* +/− heart mitochondria treated with antimycin A, a specific inhibitor that induces reactive oxygen species production from complex III, was significantly higher in comparison with the antimycin-A-treated *Cdkl5* +/+ condition (Figure 6A). The mitochondria isolated from *Cdkl5* +/− mouse hearts also showed higher levels of malondialdehyde (MDA), an end product of lipid peroxidation, consistent with a condition of oxidative stress (Figure 6B).

Interestingly, two proteins, the Poly(ADP-ribose) polymerase 1 (PARP1) and the nuclear factor erythroid 2-related factor 2 (Nrf2), that are known to be activated due to oxidative stress showed increased levels in the hearts of *Cdkl5* +/− mice compared to *Cdkl5* +/+ mice (Figure 6C–F). Similarly, we found an increase in the LC3-II/LC3-I ratio (Figure 6G,H), which is indicative of increased autophagosome formation, following oxidative stress.

## 3. Discussion

CDD is a very severe and debilitating neurodevelopmental infantile disorder with harsh neurological symptoms such as intractable seizures, neurodevelopmental delay, and autistic-like features. Nevertheless, the improvement of the clinical overview of CDD in the past few years has defined a more detailed phenotypic spectrum; this includes very common alterations in peripheral organ and tissue function, such as gastrointestinal problems, irregular breathing, hypotonia, and scoliosis [32], suggesting that CDKL5 deficiency compromises not only CNS function but also that of other organs/tissues. Here, we report, for the first time, that a mouse model of CDD, the heterozygous *Cdkl5* KO (*Cdkl5* +/−) female mouse, exhibits cardiac functional and structural abnormalities. *Cdkl5* +/− mice exhibited QTc prolongation and increased heart rate accompanied by impaired cardiac autonomic control. Moreover, the *Cdkl5* +/− heart shows typical signs of heart aging, including increased fibrosis, mitochondrial dysfunctions, and increased ROS production.

ECG measurements from *Cdkl5* +/− mice showed statistically significant increases in heart rate and in rate-corrected QTc intervals, which are indicative of delayed ventricular depolarization and repolarization. This is in line with a recent finding showing that in a cohort of individuals with CDD, the incidence of prolonged QTc or other abnormalities including sinus tachycardia was higher than the prevalence seen within the general population [26]. However, as mentioned by the authors, this first patient study has some important limitations: it was retrospective in nature, and the sample size was small [26]. Here, using an experimental approach that allows for continuous ECG recording during sleep in a validated mouse model of CDD, we overcame these limitations and, importantly, deeply reduce environmental confounders besides the genetic confounder. We found that, similarly to heterozygous *Cdkl5* +/− mice, homozygous *Cdkl5* −/− female mice exhibited tachycardia and a longer QTc interval in comparison with *Cdkl5* +/+ mice. Interestingly, the magnitude of the cardiac defects was similar in the two *Cdkl5*-deficient conditions. This is not surprising, since it has been shown that heterozygous *Cdkl5* +/− mice develop some behavioral abnormalities that are comparable to defects identified in homozygous *Cdkl5* −/− females [33]. The putative skewing toward the mutated X chromosome found in the heart of heterozygous females (levels of Cdkl5 decreased to 39%) could explain the similar cardiac phenotypic outcome show by heterozygous and homozygous *Cdkl5* KO female mice.

It is well known that neurological dysfunction may affect the control of cardiac rate and rhythm [34]. In particular, it has previously been described that autonomic neuropathies prolong QTc intervals in patients with CNS disease, including Rett syndrome [27,35,36,37,38]. Therefore, we tested the hypothesis that cardiac autonomic control is impaired in *Cdkl5* +/− mice. By applying validated indices of spontaneous RR variability in mice, we assessed cardiac vagal modulation during sleep under baseline conditions and following pharmacological manipulations. Our findings that in *Cdkl5* +/− mice, pNN8 and RMSSD indices were lower than in control mice indicate an impairment in the vagal modulation of the heart period in *Cdkl5* +/− mice. Compared to saline infusion, treatment with muscarinic antagonist (atropine) did not significantly modify the heart rate or QTc interval in *Cdkl5* +/− mice, confirming the reduction in parasympathetic tone in the heart. A growing body of evidence indicates that reduced cardiac parasympathetic activity is a common alteration in brain disorders; it is present in children with autism and Rett syndrome [39,40], as well as in neurological conditions characterized by neuronal degeneration [41,42,43]. The mechanism underlying this autonomic alteration is partly unknown [44,45]; however, reduced cardiac vagal tone is thought to be prodromic for lethal arrythmias and sudden death. The observed reduction in cardiac M_2_ receptor expression in the heart of *Cdkl5* +/− mice may be one of the possible mechanisms underlying the parasympathetic dysfunction in the mouse model of CDD. However, since, in many situations, loss of Cdkl5 function appears to lead to impaired neuronal activity [16,18,46], we cannot exclude that an overall reduction in activity in regions of the brainstem that are important for cardiorespiratory function may underlie this autonomic dysfunction. Regarding the sympathetic regulation of the heart, similarly to wild-type mice, *Cdkl5* +/− hearts responded to atenolol infusion by increasing RR and decreasing QTc intervals, indicating the presence of a preserved sympathetic modulation in the *Cdkl5* +/− heart. Interestingly, after atenolol infusion, RR and QTc intervals no longer differed between genotypes (Appendix A). Since no indices of sympathetic cardiac control have been validated for mouse heart rhythm variability to, at the moment, we cannot exclude an impairment of this autonomic branch in modulating heart rhythm in *Cdkl5* +/− mice.

The discovery that two cardiac-specific genes (*Scn5a* and *Hcn4*) are altered in their expression in *Cdkl5* +/− female hearts indicates that CDKL5, either directly or indirectly, regulates the expression of genes that play a role in the cardiac conduction systems at the cardiomyocyte level. Loss-of-function mutations in the *SCN5A* gene, which encodes the α subunit of the cardiac voltage-gated Na+ channel NaV1.5, underlie cardiac disorders, including long QT syndrome [47]. Notably, long QT has been described in almost 20% of Rett syndrome patients, and alterations in the expression of *Scn5a* genes have been reported in Mecp2-null mice [48]. The lower expression of the cardiac pacemaker-specific channel *Hcn4* found in *Cdkl5* +/− hearts could also contribute to the altered heart rate in *Cdkl5* +/− mice. Our results suggest that Cdkl5 deficiency selectively affects the expression of voltage-gated channel genes in the heart and that abnormal cardiac gene expression may be an arrhythmogenic substrate in *Cdkl5* +/− mice.

Immunostaining for Cx43 in *Cdkl5* +/− ventricular tissues revealed a significant reduction in Cx43 levels at cell–cell junctions, as confirmed by Western blot analysis. Deregulation of β-catenin levels was also present in the intercalated discs of *Cdkl5* +/− cardiomyocytes. Both Cx43, as a major component of gap junctions [49], and β-catenin, as a component of the gap junctions and intercalated discs, contribute to the regulation of the transmission of electrical signals through cardiac myocytes. Therefore, the present findings strongly suggest an impairment of intercellular communication in *Cdkl5* +/− cardiomyocytes, which could further be considered as an arrhythmogenic substrate in *Cdkl5* +/− hearts. Indeed, heterogeneous reduction in Cx43 expression and altered patterns of gap junction distribution are features of human ventricular disease and correlate with electrophysiologically identified arrhythmic changes in animal models [49,50]. It is of note that age-related disorganization of intercalated discs, including increased β-catenin expression, which may be responsible for the slower conduction of the depolarization wave within the heart, have been recently described [51], suggesting an accelerated cardiac senescence in *Cdkl5* +/− mice.

One of the main risk factors for cardiovascular diseases is aging [52]. Previously, we showed that *Cdkl5* KO mice are characterized by an increased rate of apoptotic cell death during brain aging due to accelerated neuronal senescence, a factor that causes a consequent age-related cognitive and motor decline [18]. Here, we observed a similar accelerated senescence in the heart of *Cdkl5* +/− mice. We found signs of increased cardiac fibrosis and increased extracellular matrix deposition of collagen and vimentin expression. Premature senescence develops through various external and internal stress signals, including energetic dysfunction, giving rise to free radical reactive oxygen species (ROS) that cause damage to cellular macromolecules; accumulation of this damage leads to the physiological compromise seen in aging [53]. Current evidence suggests that mitochondrial dysregulation is the cause and primary target of energetic dysfunction and free radical production [54].

In this study, we found a decrease in mitochondrial oxygen consumption rate in isolated mitochondria from *Cdkl5* +/− hearts when ADP was supplied as a substrate for the ATP synthase (state 3 respiration) and diminished specific activity of complexes I and II. These data are consistent with the decreased level of ATP and the decreased ATP/ADP ratio measured in heart tissue homogenates from *Cdkl5* +/− mice compared with wild-type animals. In addition, mitochondria isolated from the hearts of *Cdkl5* +/− mice showed increased ROS and lipid peroxidation biomarker (MDA) production compared with controls, suggesting a link between mitochondrial dysfunction and the onset of oxidative and energetic stress.

These findings are in line with recent evidence that mitochondrial dysfunction and oxidative stress occur in CDD [55,56]. Cytokine dysregulation, inflammatory status, oxidative stress marker 4HNE-Pas, and redox imbalance were evidenced in plasma from CDD patients [56,57,58]. Additionally, studies in *Cdkl5* KO mice identified brain mitochondrial functional abnormalities, including reduced activity of mitochondrial respiratory chain complexes and impairment in mitochondrial ATP production rate [59,60], as well as alterations in patient-derived iPSCs [61,62]. It is worth noting that we found mitochondrial structural changes that were consistent with the development of intramitochondrial crystalline-like inclusions in the *Cdkl5* +/− heart. The presence of mitochondrial paracrystalline inclusions in a clinically characterized group of patients with genetically defined mitochondrial disease [63,64] suggests that this ultrastructural alteration could contribute to mitochondrial function. Therefore, we can speculate that the crystalline inclusions found in the inner membrane may underlie the bioenergetic defects found in *Cdkl5* +/− mitochondria. However, we cannot exclude the possibility that the crystalline-like inclusions represent compensatory responses to mitochondrial stress.

Recent evidence has described a potential role of AKT and autophagy, molecular mechanisms of cellular senescence, in aging-associated organ deterioration. When cardiomyocytes incur oxidative stress, they activate the PI3K/AKT signaling pathway [65], suggesting that oxidative stress is an upstream event related to the activation of the AKT pathway in cardiomyocytes. Our finding that AKT pathway activation is increased in the hearts of *Cdkl5* +/− female mice could be a consequence of an increase in the generation of ROS due to aberrant mitochondrial function. Similarly, increased LC3-II levels, which are associated with either enhanced autophagosome synthesis or reduced autophagosome turnover [66], may be associated with increased ROS production and the resulting oxidative cell stress that occurs in many disease states [67]. We found that the ratio of LC3-II/LC3-I was enhanced in the *Cdkl5* +/− heart, suggesting an increasing number of autophagosomes.

The principal growth-promoting intracellular signaling pathways that are activated by ROS in cardiac myocytes include not only the AKT pathway but also the mitogen-activated protein kinase cascades (ERK1/2 pathway) [65]. In association with *Cdkl5* +/− cardiac myocyte dysfunctions, we found alterations in AKT activity but not in ERK1/2 signaling. Activation of ERK1/2 is generally associated with cell growth and survival, and studies of transgenic mice have shown that selective activation of the ERK1/2 cascade in the myocardium induces adaptive cardiac hypertrophy [68]. Further evidence linking ERK1/2 with hypertrophy was reviewed in [68]. Since no signs of hypertrophy are evident in the *Cdkl5* +/− heart, this might explain the lack of ERK1/2 dysregulation in these mice.

Considering the protective molecular pathways that might be triggered by oxidative stress [69] and autophagy [70], we found increased levels of Nrf2, a transcription factor known to activate multiple enzymes with antioxidant properties [71], as well as increased levels of PARP1, a DNA damage sensor that facilities base excision repair [72] in the *Cdkl5* +/− heart. We hypothesized that increased levels of Nrf2 are triggered by ROS production and that this increase is most likely a compensatory effort to increase antioxidant defenses in the *Cdkl5* +/− context. This should correlate with a reduction in oxidative stress in *Cdkl5* +/− hearts; however, this is not the case, and the issue requires further investigation. It is worth noting that PARP activation has been demonstrated to impair mitochondrial function [73] and promote autophagy in cardiomyocytes [74]. Therefore, increased PARP1 levels may have contributed to the pathologic signs observed in the *Cdkl5 +/*− heart.

At present, we have no evidence for which of the observed Cdkl5-related alterations is the primary cause underlying the structural and functional abnormalities in the *Cdkl5* +/− heart. However, increasing evidence suggests that ROS production is associated with cardiac arrhythmias; in particular, elevated cellular ROS can cause alterations of the cardiac ion channels, changes in mitochondrial function, and gap junction remodeling, leading to arrhythmic conditions [75,76,77]. Therefore, oxidative stress, a common pathophysiological factor in cardiac disease [78,79,80], may be the main defect underlying cardiac alterations in the *Cdkl5*-deficient heart. Future studies aimed at rescuing the cardiac phenotype in *Cdkl5* KO mice, targeting one of the observed defects, could help to shed light on the primary causes that drive cardiac abnormalities in the absence of *Cdkl5*.

## 4. Materials and Methods

### 4.1. Animal Husbandry

The mice used in this work were derived from the *Cdkl5* −/Y strain in the C57BL/6N background developed in [8] and backcrossed in C57BL/6J for three generations. Heterozygous *Cdkl*5 +/− and homozygous *Cdkl5* −/− females were produced and genotyped as previously described [8], and age-matched wild-type *Cdkl5* +/+ littermate controls were used for all experiments. The day of birth was designated as postnatal day (P) zero, and animals of 24 h of age were considered as 1-day-old animals (P1). After weaning (P21-23), mice were housed three to five per cage with a 12 h light/dark cycle in a temperature- and humidity-controlled environment with food and water provided ad libitum. The animals’ health and comfort were controlled by the veterinary service. Experiments were carried out on a total of 121 adult (3–4-month-old) *Cdkl5* KO mice (*Cdkl5* +/+ n = 50; *Cdkl5* +/− n = 62; *Cdkl5* −/− n = 9). The study protocols complied with EU Directive 2010/63/EU and with Italian law (DL 26, 4 March 2014) and were approved by the Italian Ministry of Health (protocol n° 535/2022-PR). All efforts were made to minimize animal suffering and to keep the number of animals used to a minimum.

### 4.2. Surgical Procedure, In Vivo Recording, and Data Analysis

Eight *Cdkl5* +/+, 10 *Cdkl5* +/−, and 8 *Cdkl5* −/− female mice were instrumented with electroencephalographic (EEG), electromyographic (EMG), and electrocardiographic (ECG) electrodes to characterize the electrical activity of the heart during sleep under baseline conditions. A second group of mice (*Cdkl5* +/+ n = 10, *Cdkl5* +/− n = 10) instrumented with the same surgical protocol also underwent surgery to implant an intraperitoneal (IP) catheter for the continuous infusion of autonomic blockers with the purpose of testing the autonomic modulation of heart rhythm.

All mice were deeply anesthetized with isoflurane (1.8–2.4% in O_2_, inhalation route) and treated with intraoperative analgesia (carprofen 4 mg/kg subcutaneously, Pfizer, Italy) and postoperative antibiotic prophylaxis (benzylpenicillin benzathine, 12,500 IU/kg, and dihydrostreptomycin sulphate, 5 mg/kg, subcutaneously). For the recording of the EEG signal, mice were implanted with two miniature stainless-steel screws (2.4 mm length, PlasticsOne, Roanoke, VA, USA) in contact with the dura mater (frontoparietal derivation); for the recording of the EMG signal, mice were implanted with two multistranded PFA-coated stainless-steel wires (KF Technology srl, Roma, Italy) inserted into the nuchal muscles. For ECG recording, two PFA-coated stainless-steel wires were inserted subcutaneously; one was put in contact with muscles in the right-upper quadrant of the thorax, while the second touched the abdominal muscles on the left flank. All electrode wires were then collected in a socket placed over the mouse’s head and fixed with dental cement (RelyX Unicem, 3M ESPE, Pioltello, (MI), Italy) and dental acrylic (Respal NF, SPD, Mulazzano (LO), Italy). The second batch of mice was also implanted with a silicone catheter [81,82] with the tip inserted into the abdominal cavity and the other extremity tunneled to the mouse head and fixed with the abovementioned socket. After 12 days of postoperative recovery and habituation to the recording apparatus (ambient temperature set at 25 °C), EEG, EMG, and ECG signals of mice included in the baseline protocol were continuously recorded for 24 h. Signals were transmitted via a cable connected to a rotating electrical commutator (SL2 + 2C/SB, Plastics One, USA) and to a balanced cable suspensor, allowing the mice to make unhindered movements [83].

Mice included in the protocol for testing of the autonomic control of heart rhythm were allowed to recover from surgery and to habituate to the recording settings for 14 days. Each mouse underwent 3 recording sessions (7 h each starting at light on) with acquisition of EEG, EMG, and ECG signals while being continuously infused with either saline, with the muscarinic receptor antagonist atropine methyl nitrate (Vinci-Biochem, Italy, 0.5 mg/mL in saline) to block parasympathetic activity to the heart, or with the selective β1-adrenergic receptor antagonist atenolol (Sigma-Aldrich, Milano, Italy, 0.25 mg/mL in saline) to block the sympathetic activity to the heart. Each mouse first received saline infusion and was then randomly subjected to atropine and atenolol infusions. Each infusion was performed at least 48 h after the preceding infusion. The IP catheter was connected to a remote infusion pump (model 22 multiple syringe pump, Harvard Apparatus, Cambridge, MA, USA) by an external tube prefilled with saline solution or drug solution, as previously described [81,82]. A rapid infusion was performed before the start of each recording session at a rate of 30 μL/min for 5 min to fill the IP catheter with saline or either of the drug solutions; then, the infusion rate was set at 100 μL/h for 7 h. At the end of the recording session, mice were sacrificed under deep anesthesia (isoflurane 4% in O_2_).

The EEG signal was cut with a band pass filter between 0.3 and 100 Hz and stored at 128 Hz. The EMG signal was filtered between 100 and 1000 Hz and stored at 128 Hz, while the ECG was filtered between 10 and 1000 Hz and stored at 2048 Hz. Data acquisition was performed with LabVIEW 8.0 software (National Instruments, Austin, TX, USA).

The EEG and EMG signals were imported and analyzed to automatically score the wake–sleep states (wakefulness, rapid-eye-movement sleep (REMS) and non-rapid-eye-movement sleep (NREMS)). On the contrary, the ECG signal was imported in Labchart 8.0 (ADInstruments, Colorado Springs, CO, USA), a specific notch (50 Hz) filter was digitally applied to exclude electrical noise, and the signal was then analyzed with the ECG analysis module to automatically detect QRS complexes and their intervals. For each mouse, ECG analysis was restricted to 8 episodes of NREMS and 8 episodes of REMS (longer than 30 s) homogeneously distributed throughout the whole recording. For each session, we calculated the following ECG parameters: P-wave duration, QRS, RR, PR, JT, and QT intervals. To compensate for the elevated mouse heart rate, we corrected the QT interval (QTc) with Hodges’ formula (the most effective for rodents, [84]).

The vagal (parasympathetic) contribution to cardiac modulation was examined using validated mouse indices computed in the time domain on spontaneous RR fluctuations [85]: pNN8, % of RR values that differ from the following values by >8 ms [86]; RMSSD, root mean square of the successive RR differences [87]. To date, no index of spontaneous RR fluctuation in the time domain has been validated for the sympathetic contribution to cardiac modulation.

### 4.3. Heart Dissection, Measurement, and Collection

Adult *Cdkl5* −/−, *Cdkl5* +/−, and *Cdkl5* +/+ female mice aged 3–4 months were weighed and put under deep anesthesia through inhalation of 2% isoflurane in pure oxygen and sacrificed through cervical dislocation. Hearts were quickly removed, cleaned from the surrounding structures, and thoroughly washed in PBS to remove all blood, then weighed. The ratio of heart weight to body weight (HW/BW) was then calculated by dividing the weight of the heart by the weight of the whole animal. The atrioventricular distance was measured. All measurements were performed by the same person with the same precision scales. Hearts were quickly frozen in isopentane, cooled in liquid nitrogen, and stored at −80 °C until used for RT-qPCR, immunohistochemistry, and Western blot analyses.

### 4.4. RNA Isolation and RT-qPCR

RNA isolation and RT-qPCR were conducted on frozen hearts of *Cdkl*5 −/−, *Cdkl*5 +/−, and *Cdkl*5 +/+ female mice and on frozen cortices of *Cdkl*5 +/+ female mice. Total RNA was isolated using the TRI reagent method (Sigma-Aldrich, St. Louis, MO, USA), and cDNA synthesis was achieved with 5 μg of total RNA using an iScript™ advanced cDNA synthesis kit (Bio-Rad, Hercules, CA, USA) according to the manufacturer’s instructions. Real-time PCR was performed using SsoAdvanced Universal SYBR Green Supermix (Bio-Rad) in a CFX real-time PCR detection system (Bio-Rad). We used primer pairs (Appendix A) that provided an efficiency close to 100%. Each biological replicate was run in triplicate. The mean of the two more stable reference genes (*Actb* and *Gapdh*) were used as a normalization factor in the RT-qPCR analysis, and relative quantification was performed using the ΔΔCt method.

### 4.5. Histological and Immunohistochemistry Procedures

For histological and immunohistochemistry procedures, frozen hearts were cut with a cryostat (Histo-Line Laboratories) into 7 μm thick sections. Sections were mounted on super frost slides.

#### 4.5.1. Hematoxylin and Eosin (H&E)

Sections were stained with hematoxylin and eosin (H&E), immersed in graded alcohols followed by xylene, and mounted in mounting medium (DPX mountant, Sigma-Aldrich, St. Louis, MO, USA).

#### 4.5.2. Masson’s Trichrome Staining

Sections were stained with Masson’s trichrome (MT) using Masson’s Trichrome Stain Kit-Mallory’s (StatLab, KTMAL); the procedure was carried out according to the protocol included with the kit.

#### 4.5.3. Connexin 43, Actinin, and β-Catenin Staining

Immunohistochemistry was performed on 7 μm thick frozen sections fixed via immersion in 4% paraformaldehyde (100 mM phosphate buffer, pH 7.4) and permeabilized with 0.2% TritonX-100 in PBS. Furthermore, 2% BSA in PBS was used as a blocking reagent. Sections were incubated overnight with anti-β-Catenin antibody or double-stained with anti-connexin 43 and anti-actinin antibodies, washed with PBS, and subsequently incubated for 2 h at room temperature with FITC-conjugated or CY3-conjugated secondary antibodies. The primary and secondary antibodies used in this study are listed in Appendix A. The sections were mounted with DAPI (4′,6-diamidino-2-phenylindole)-Fluoromount-G (SouthernBiotech, AL).

### 4.6. Image Acquisition and Measurements

Fluorescence images were taken with an Eclipse TE 2000-S microscope equipped with a DS-Qi2 digital SLR camera (Nikon Instruments Inc.). A light microscope (Leica Mycrosystems) equipped with a motorized stage and focus control system and a color digital camera (Coolsnap-Pro, Media Cybernetics) were used to take brightfield images of Masson’s trichrome and hematoxylin eosin-stained sections.

#### Quantification of β-Catenin Staining Intensity and Areas

Starting from 20× magnification images of β-catenin-stained ventricular slices, the area of β-catenin staining in intercalated discs was manually drawn using the Image Pro Plus measurement function and expressed in μm^2^. The intensity of β-catenin staining within each area was then quantified by determining the sum intensity of all positive (bright) pixels within the area. Approximately 100 intercalated discs were analyzed from each sample.

### 4.7. Western Blotting

For the preparation of protein extracts, ventricles were homogenized in RIPA buffer and quantified using the Bradford method as previously described [88].

Equivalent amounts (50 μg) of protein were subjected to electrophoresis on a 4–12% Mini PROTEAN^®^ TGX™ Gel (Bio-Rad) and transferred to a Hybond ECL nitrocellulose membrane (GE Healthcare Bio-Science). The primary and secondary antibodies used are listed in Appendix A. The densitometric analysis of digitized Western blot images was performed using Chemidoc XRS Imaging Systems and Image Lab^TM^ software (Bio-Rad), which automatically highlights any saturated pixels of the Western blot images in red. Images acquired with exposition times that generated protein signals outside of a linear range were not considered for quantification.

### 4.8. Adenine Nucleotide Measurement

For the determination of adenine nucleotides, mouse hearts were quickly removed and placed in ice-cold PBS supplemented with 10 mM EDTA. The tissue was homogenized using an Ultra-Turrax (1 cycle for 10 s); then, the resuspension was transferred to a glass potter with a Teflon pestle operated at 1600 r.p.m (10 strokes). All steps were performed at 4 °C. ATP and ADP were extracted and detected as previously described [89] using HPLC (Agilent 1100 series system) using a Kinetex C18 column (250 × 4.6 mm, 100 Å, 5 μm; Phenomenex). Nucleotide peaks were identified at λ = 260 nm by comparison and coelution with the standards. Different nucleotides were quantified through peak area measurement compared with standard curves.

### 4.9. Mitochondrial Oxygen Consumption

Intact mitochondria from mice hearts were isolated as previously described [90]. Briefly, freshly prepared mitochondria were assayed for oxygen consumption at 30 °C by means of a thermostatically controlled oxygraph apparatus (Instech Mod. 203, Plymouth Meeting, PA, USA) equipped with a Clark electrode and a rapid mixing device. Mitochondria (0.2 mg protein) were incubated in the respiration medium (10 mM Tris/HCl, 5 mM MgCl_2_, 2 mM Pi, 20 µM EGTA and 0.25 M sucrose, pH 7.4) in a final volume of 1600 µL. Respiratory substrates (5 mM glutamate–malate or 13 mM succinate) were added after signal stabilization. State 4 respiration was recorded for 2 min, and state 3 respiration was induced by the addition of 300 µM ADP. The respiratory rates were expressed in nmol oxygen/min/mg of protein.

### 4.10. Enzymatic Activities of the Mitochondrial Respiratory Chain

The activity of mitochondrial complexes I and II was assessed in heart mitochondria isolated from *Cdkl*5 +/− and *Cdkl*5 +/+ mice as previously described [91] with minor modifications. Complex I (NADH: ubiquinone oxidoreductase) activity was measured spectrophotometrically following NADH (ε = 6.22 mM cm^−1^) oxidation at λ = 340 nm and 37 °C using a Jasco-v750 spectrophotometer equipped with a cuvette stirring device and thermostatic control. Briefly, the isolated mitochondria were subjected to three freeze–thaw cycles in hypotonic buffer (20 mM phosphate buffer (pH 7.5)) to avoid compartmentalization. Then, 50 µg of mitochondria was added to 700 µL of distilled water in a 1 mL quartz cuvette for 1 min. Then, 100 μL of potassium phosphate buffer (0.5 M, pH 7.5), 60 μL of fatty-acid-free BSA (50 mg ml^−1^), 30 μL of KCN (10 mM), and 10 μL of NADH (10 mM) were added. The volume was adjusted to 994 µL with distilled water. In parallel, a separate cuvette containing the same quantity of reagents and sample with the addition of 10 μL of 1 mM rotenone was prepared. The reaction was started by adding 6 µL of 10 mM CoQ1 (2,3-dimethoxy-5-methyl-6-(3-methyl-2-butenyl)-1,4-benzoquinone, Sigma-Aldrich), and the decrease in absorbance at 340 nm was followed for 3 min. Specific complex I activity was obtained by subtracting the rotenone-insensitive activity and normalized to mitochondrial protein content as determined by a Lowry assay [92]. Complex II (succinate dehydrogenase) was measured spectrophotometrically at λ = 600 nm following the reduction of DCPIP (2,6-dichlorophenolindophenol, Sigma-Aldrich) (ε = 19.1 mM cm^−1^) at 37 °C. Briefly, 60 µg of cell lysate, 50 μL of potassium phosphate buffer (0.5 M, pH 7.5), 20 μL of fatty-acid-free BSA (50 mg ml^−1^), 30 μL of 10mM KCN, 50 μL of 400 mM succinate, and 145 µL of DCPIP (0.015% (wt/vol) were added to a 1 mL glass cuvette, and the volume was adjusted to 996 µL with distilled water. After ten minutes of incubation inside the spectrophotometer at 37 °C, the reaction was started by adding 4 μL of 12.5 mM decylubiquinone (2,3-dimethoxy-5-methyl-6-decyl-1,4-benzoquinone, Sigma-Aldrich), and the absorbance decrease at λ = 600 nm was followed for 3 min. To check the specificity of complex II activity, 10 µL of 1 mM carboxine (5,6-dihydro-2-methyl-N-phenyl-1,4-oxathiin-3-carboxamide, Carbathiine, Sigma-Aldrich) was added before starting the reaction. The specific activity of complex II was obtained by subtracting the TTFA-insensitive activity and normalized to mitochondrial protein content determined by a Lowry assay [92].

### 4.11. Measurement of ROS Production

Reactive oxygen species production was assessed in cardiac mitochondria from *Cdkl*5 +/− and *Cdkl*5 +/+ female mice using a fluorogenic probe (2′,7′-dichlorodihydrofluorescein diacetate; Thermo Fisher Scientific Inc, Waltham, MA, USA) as previously described [93]. For ROS determination, mitochondria (0.1 mg/mL in 125 mM KCl, 10 mM TRIS, 1 mM EDTA, pH 7.5) were incubated with 2 mM KCN and 10 µM DCFDA for 10 min in the presence and absence of 2 µM antimycin A. After this time, mitochondria were energized with 15 mM succinate, and the DCF fluorescence emission (λ_ex_ = 485 nm; λ_em_ = 535 nm) was recorded 10 min after energization using a plate reader (EnSpire; PerkinElmer).

### 4.12. Measurement of Lipid Peroxidation

Lipid peroxidation in isolated mitochondria from *Cdkl*5 +/− and *Cdkl*5 +/+ female mice was assessed by measuring the biomarker malondialdehyde (MDA) as previously described by Reilly et al. [94]. Briefly, quantification of MDA was performed through reaction with thiobarbituric acid (TBA), and measurement of TBA-MDA adduct was carried out at 535 nm using a Jasco V-750 spectrophotometer. 1,1,3,3-tetramethoxypropane (Sigma-Aldrich, St. Louis, MO, USA) was used as a standard.

### 4.13. Transmission Electron Microscopy

For ultrastructural characterization, small heart specimens were fixed in 2.5% glutaraldehyde in 0.1 M cacodylate buffer and post-fixed in osmium tetroxide 1% in the same buffer. After dehydration in graded ethanol, specimens were embedded in araldite. Thin sections were counterstained with uranyl acetate and lead citrate and observed under a Philips CM100 transmission electron microscope (Philips, Amsterdam, the Netherlands).

### 4.14. Statistical Analysis

Statistical analysis was performed using GraphPad Prism (version 9). Values are expressed as means ± standard error (SEM). The significance of results was obtained using a two-tailed Student’s *t*-test or one-way and two-way ANOVA, followed by Fisher’s LSD post hoc test, as specified in the figure legends. For the in vivo recordings under the baseline condition, data were analyzed with 2-way repeated-measure ANOVAs (considering sleep state and genotype as factors) and corrected for multiple comparisons with Sidak’s test. For mice included in the pharmacological testing of autonomic cardiac contribution, we first calculated the % modification of RR and QTc values induced by the drug compared to saline infusion. Then, for each experimental group, we performed a one-sample t-test, checking for significant deviations from the reference value (100%). A probability level of *p* < 0.05 was considered to statistically significant. The confidence level was taken as 95%.

## 5. Conclusions

In conclusion, the present study shows that *Cdkl5* deficiency impacts heart structure and function in mice and that *Cdkl5 +/*− mice could be a precious asset to accelerate the comprehension of these aspects. As our work does not provide direct causal evidence linking local CDKL5 expression in the heart or deregulation of direct CDKL5 phosphorylation targets to the electrical activity and structure of the cardiomyocyte, further in vitro and in vivo studies of the effects of specific knockout/rescuing CDKL5 expression in cardiomyocytes are needed. However, the results obtained in the present study could have a considerable impact in promoting a more in-depth investigation of cardiac function in patients for a better understanding of the CDD phenotypes. Although it is known that protein kinases can control cellular signaling by interacting with many targets, our knowledge of the CDKL5 phosphorylation targets is still very limited, hindering the investigation of direct CDKL5-dependent mechanisms underlying cardiac alterations in *Cdkl5* +/− mice. However, many alterations found in the heart of *Cdkl5* +/− mice are also present in the RTT mouse model, Mecp2-null mice [48,95,96], and/or in RTT patients [97,98,99], suggesting that similar molecular alterations may underlie the cardiac dysfunctions in these conditions, with several overlapping phenotypic features. Evidence that CDKL5 phosphorylates MeCP2 in vitro [100,101] suggests that MeCP2-dependent transcription regulation may be influenced by CDKL5. However, due to conflicting results [102,103], the molecular relationship between CDKL5 and MeCP2 remains to be clarified. A future in-depth gene expression analysis in CDKL5 and MeCP2-null conditions could be crucial in order to clarify the similarities and differences between CDD and RTT in terms of their cardiac phenotypes.

## Figures and Tables

**Figure 1 ijms-24-05552-f001:**
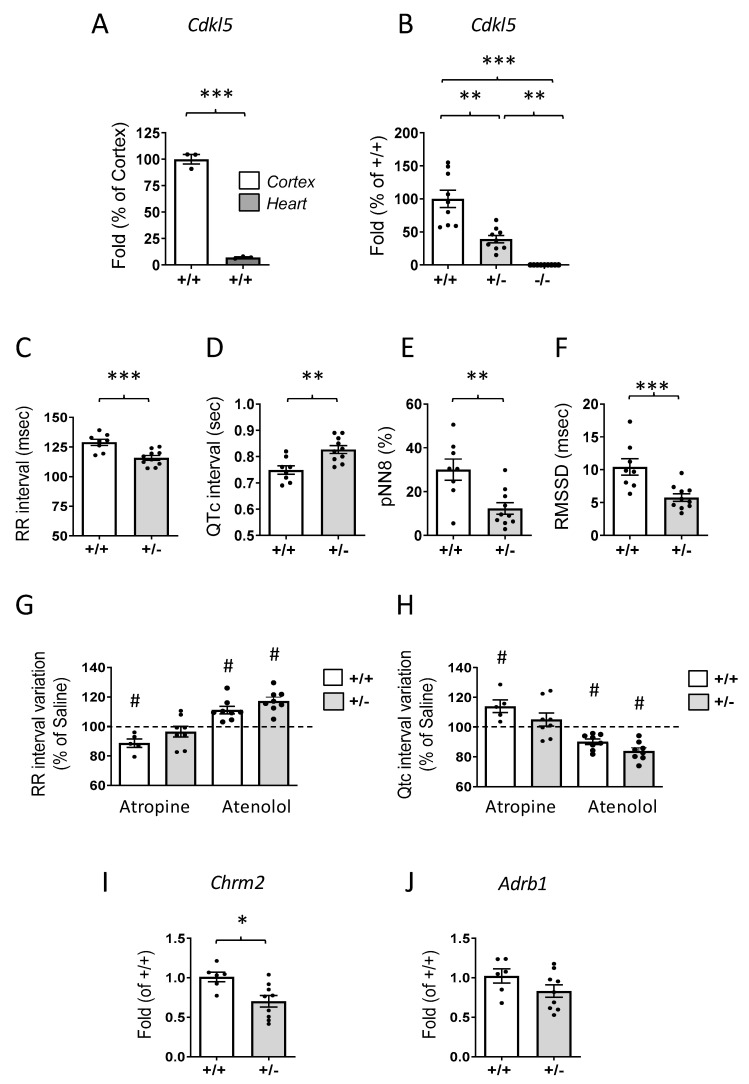
***Cdkl5* expression in the heart of *Cdkl5* +/− mice and ECG analysis.** (**A**) Real-time qPCR analysis of *Cdkl5* mRNA expression in the cortex (n = 3) and heart (n = 3) of 3-month-old wild-type *Cdkl5* +/+ mice. Data are presented as a percentage of *Cdkl5* cortical expression. *** *p* < 0.001 (two-tailed Student’s *t*-test). (**B**) Relative *Cdkl5* mRNA expression in the heart of 3-month-old *Cdkl5* +/+ (n = 6) heterozygous *Cdkl*5 +/− (n = 9) and homozygous *Cdkl5* −/− (n = 9) female mice. The results are expressed as percentages of *Cdkl5* cardiac expression in *Cdkl5* +/+ mice. ** *p* < 0.01; *** *p* < 0.001 (Fisher’s LSD test after one-way ANOVA). (**C**) Mean heart period duration (expressed as intervals between two consecutive R waves) during non-rapid-eye-movement sleep (NREMS) in *Cdkl5 +/+* (n = 8) and *Cdkl5 +/*− (n = 10) female mice. (**D**) Mean duration of ventricular depolarization and repolarization (interval between Q and T waves) during NREMS. These values are reported as corrected QT intervals (QTc) after applying Hodge’s formula to consider potential differences in RR values between groups (*Cdkl5 +/+* (n = 8) and *Cdkl5 +/*− (n = 10) female mice). (**E**) Percentage of RR values that differ from the following values by more than 8 ms (pNN8) during NREMS in *Cdkl5 +/+* (n = 8) and *Cdkl5 +/*− (n = 10) female mice. (**F**) Root mean square of the successive RR differences (RMSSD) during NREMS in *Cdkl5 +/+* (n = 8) and *Cdkl5 +/*− (n = 10) female mice. The results in (**C**–**F**) are presented as means ± SEM. ** *p* < 0.01, *** *p* < 0.001 (two-tailed Student’s t-test after two-way ANOVA). (**G**) Percentage modification of the RR interval induced by either atropine (5 *Cdkl5 +/+* and 8 *Cdkl5 +/*− mice) or atenolol (8 *Cdkl5 +/+* and 8 *Cdkl5 +/*− mice) infusion compared to saline infusion (horizontal dotted line). (**H**) Percentage modification of the QTc interval induced by either atropine (5 *Cdkl5 +/+* and 8 *Cdkl5 +/*− mice) or atenolol (8 *Cdkl5 +/+* and 8 *Cdkl5 +/*− mice) infusion compared to saline infusion (horizontal dotted line). The results in (**G**,**H**) are presented as means ± SEM. # < 0.05 vs. saline infusion reference value (dotted line, one-sample *t*-test). (**I**,**J**) Real-time qPCR analysis of the muscarinic acetylcholine receptors 2 ((**I**), *Chrm2*) and the adrenoceptor beta 1 ((**J**), *Adrb1*) gene expression in the ventricle of 3-month-old *Cdkl5* +/+ (n = 6) and *Cdkl5* +/− (n = 9) mice. Data are presented as fold change in comparison with ventricular tissue from *Cdkl5* +/+ mice. * *p* < 0.05 (two-tailed Student’s *t*-test).

**Figure 2 ijms-24-05552-f002:**
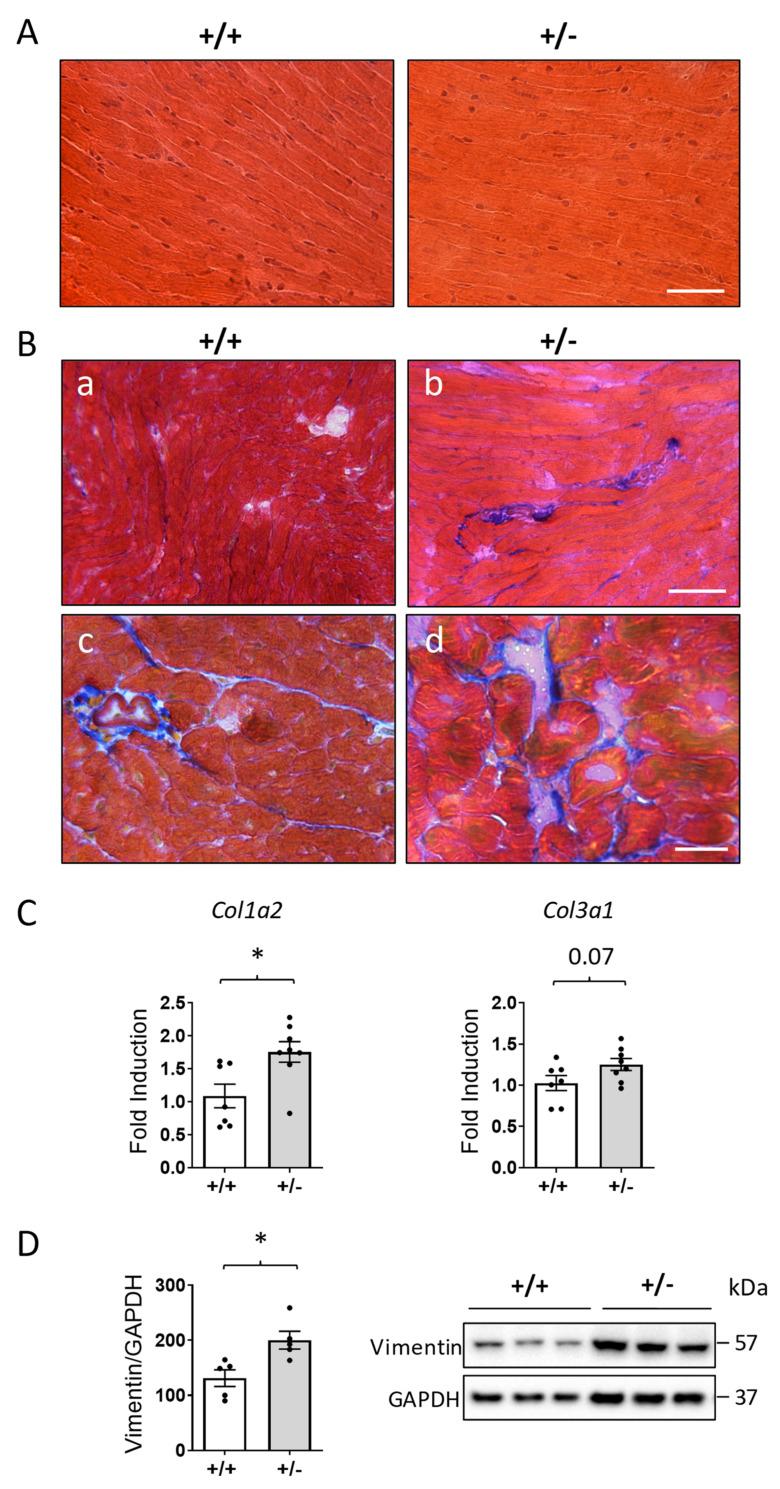
**Myocardial fibrosis in the heart of *Cdkl5* +/− mice.** (**A**) Representative images of hematoxylin and eosin staining of longitudinal sections of the ventricular tissue of *Cdkl5* +/+ (left panel) and *Cdkl5* +/− (right panel) mice. Scale bar = 50 μm. (**B**) Representative images of Masson’s trichrome staining of longitudinal (**a**,**b**) and transverse (**c**,**d**) sections of the ventricular tissue of *Cdkl5* +/+ (left panels) and *Cdkl5* +/− (right panels) mice showing an increase in blue-stained collagen fibers in the heart of *Cdkl5* +/− mice compared to wild-type (+/+) mice. Scale bars = 50 μm. (**C**) Expression of collagen type I alpha 2 chain (*Col1a2*) and collagen type III alpha 1 chain (*Col3a1*) in ventricular tissue isolated from the heart of 3-month-old *Cdkl5* +/+ (n = 7) and *Cdkl5* +/− (n = 8) mice. Data are presented as fold change in comparison with cardiac tissue from *Cdkl5* +/+ mice. (**D**) Histogram showing quantification of vimentin protein levels normalized to GAPDH in protein extracts of ventricular tissue from 3-month-old *Cdkl5* +/+ (n = 5) and *Cdkl5* +/− (n = 5) mice. Data are expressed as percentages of *Cdkl5* +/+ mice. Example of immunoblots of three animals from each experimental group on the right. Values in (**C**,**D**) are presented as means ± SEM; * *p* < 0.05 (two-tailed Student’s *t*-test).

**Figure 3 ijms-24-05552-f003:**
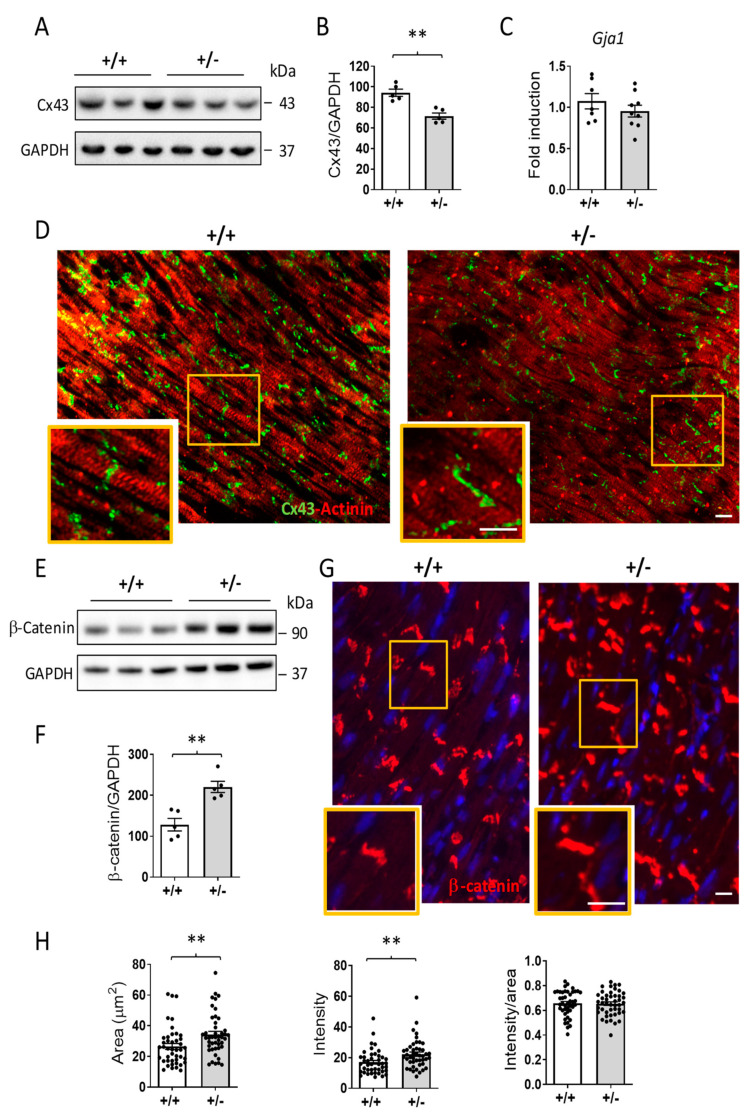
**Connexin 43 and β-catenin expression in the heart of *Cdkl5* +/− mice.** (**A**) Example of an immunoblot showing connexin 43 (Cx43) and GAPDH levels in extracts of ventricular heart tissue of *Cdkl5* +/+ (n = 3) and *Cdkl5* +/− (n = 3) mice. (**B**) Histogram showing quantification of connexin 43 (Cx43) protein levels normalized to GAPDH in extracts of ventricular heart tissue from *Cdkl5* +/+ (n = 5) and *Cdkl5* +/− (n = 5) mice. Data are expressed as percentages of *Cdkl5* +/+ mice. (**C**) Relative gap junction protein alpha 1 (*Gja1*) mRNA expression in ventricular tissue isolated from the heart of 3-month-old *Cdkl5* +/+ (n = 6) and *Cdkl5* +/− (n = 9) mice. Data are presented as fold change in comparison with cardiac tissue from *Cdkl5* +/+ mice. (**D**) Representative fluorescence images of ventricular tissue sections immunostained for connexin 43 (green) and actinin (red) from *Cdkl5* +/+ (left panel) and *Cdkl5* +/− (right panel) mice. The orange boxes highlight the regions shown in the high-magnification panels. Scale bars = 50 μm (low and high magnifications). (**E**,**F**): Western blot analysis of β-catenin levels normalized to GAPDH levels in protein extracts of ventricular heart tissue of *Cdkl5* +/+ (n = 5) and *Cdkl5* +/− (n = 5) mice. Examples of immunoblotting in (**E**) and quantification in (**F**). Data are presented as a percentage of β-catenin levels in *Cdkl5* +/+ mice. (**G**) Representative fluorescent images of ventricular tissue sections immunostained for β-catenin (red) from *Cdkl5* +/+ (left panel) and *Cdkl5* +/− (right panel) mice. Nuclei were counterstained with DAPI. The orange boxes highlight the regions shown in the high-magnification panels. Scale bars = 50 μm (low and high magnifications). (**H**) Quantification of the area (left histogram), intensity (middle histogram), and intensity per area (right histogram) of β-catenin immunostaining in ventricular tissue sections from *Cdkl5* +/+ and *Cdkl5* +/− mice. The results in (**B**,**C**,**F**,**H**) are presented as means ± SEM. ** *p* < 0.01 (two-tailed Student’s *t*-test).

**Figure 4 ijms-24-05552-f004:**
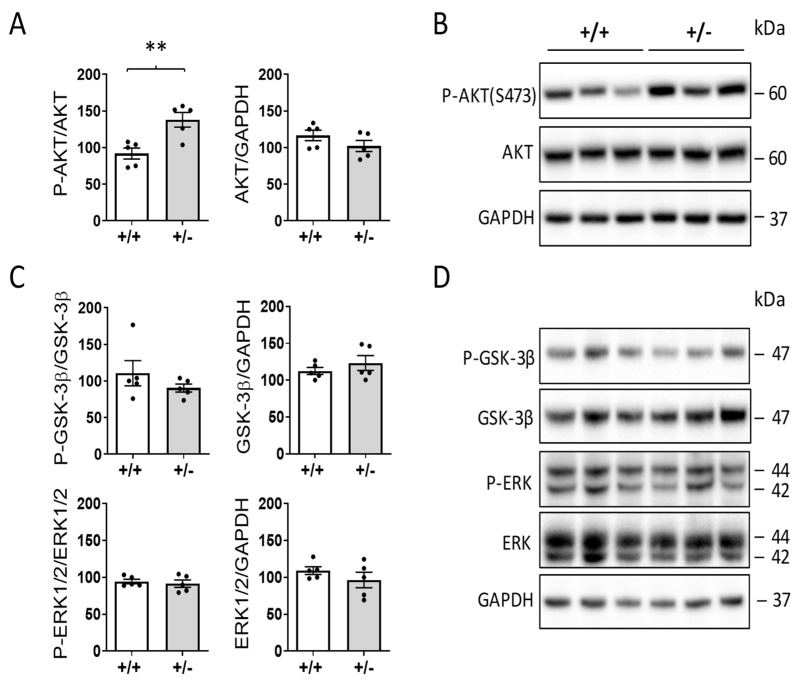
**AKT, GSK-3β, and ERK pathways in the hearts of *Cdkl5* +/− mice.** (**A**) Western blot analysis of phosphorylated AKT (P-AKT, Ser473) levels normalized to total AKT content (left histogram) and total AKT levels normalized to GAPDH levels (right histogram) in ventricular extracts from *Cdkl5* +/+ (n = 5) and *Cdkl5* +/− (n = 5) mice. (**B**) Examples of immunoblots for P-AKT, AKT, and GAPDH for three mice from each experiential group. (**C**) Western blot analysis of P-GSK-3β and P-ERK1/2 levels in ventricular extracts of mice as in (**A**). Histograms on the left show P-GSK-3β (upper) and P-ERK1/2 (lower) levels normalized to corresponding total protein content. Histograms on the right show ERK1/2 and GSK-3β protein levels normalized to GAPDH levels. (**D**) Examples of immunoblots for P-GSK-3β, GSK-3β, P-ERK1/2, ERK, and GAPDH of three animals from each experimental group. Data are expressed as a percentage of expression in *Cdkl5* +/+ mice. Values represent mean ± SEM. ** *p* < 0.01 (two-tailed Student’s *t*-test).

**Figure 5 ijms-24-05552-f005:**
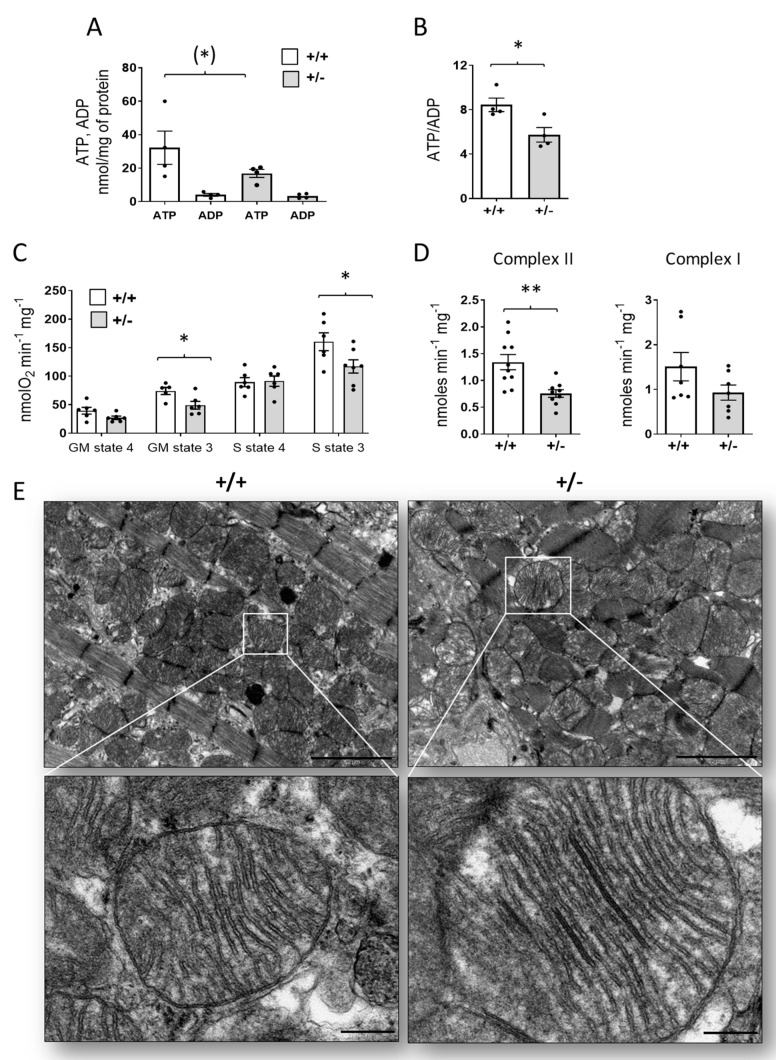
**Mitochondrial activity in the hearts of *Cdkl5* +/− mice**. (**A**,**B**) ATP and ADP content (**A**) and ATP/ADP ratio (**B**) in heart homogenates from *Cdkl5* +/+ and *Cdkl5* +/− mice measured by HPLC (n = 4). (**C**) Oxygen consumption rate (OCR) in isolated cardiac mitochondria from *Cdkl5* +/+ and *Cdkl5* +/− mice. The mitochondria were energized with glutamate–malate (GM) or succinate (S) in the presence (state 3) and absence (state 4) of ADP (n = 6). (**D**) Specific complex I (NADH-CoQ_1_) and complex II (succinate-DCPIP) activities in isolated cardiac mitochondria from *Cdkl5* +/+ and *Cdkl5* +/− mice (n = 10). (**E**) Representative electron microscopic images of ventricular tissue from *Cdkl5* +/+ (left panel) and *Cdkl5* +/− (right panel) mice. In the *Cdkl5* +/− heart, linear densities are evident inside the intracristal matrix of mitochondria, while a normal cristae structure is observed in *Cdkl5* +/+ mitochondria. The white boxes highlight the regions shown in the high-magnification panels. Scale bars: low magnifications = 2 μm, high magnifications = 200 nm. The results in (**A**–**D**) are presented as means ± SEM; in (**A**), ^(^*^)^
*p* = 0.055 (Fisher’s LSD test after two-way ANOVA); in (**B**–**D**), * *p* < 0.05, ** *p* < 0.01 (two-tailed Student’s *t*-test).

**Figure 6 ijms-24-05552-f006:**
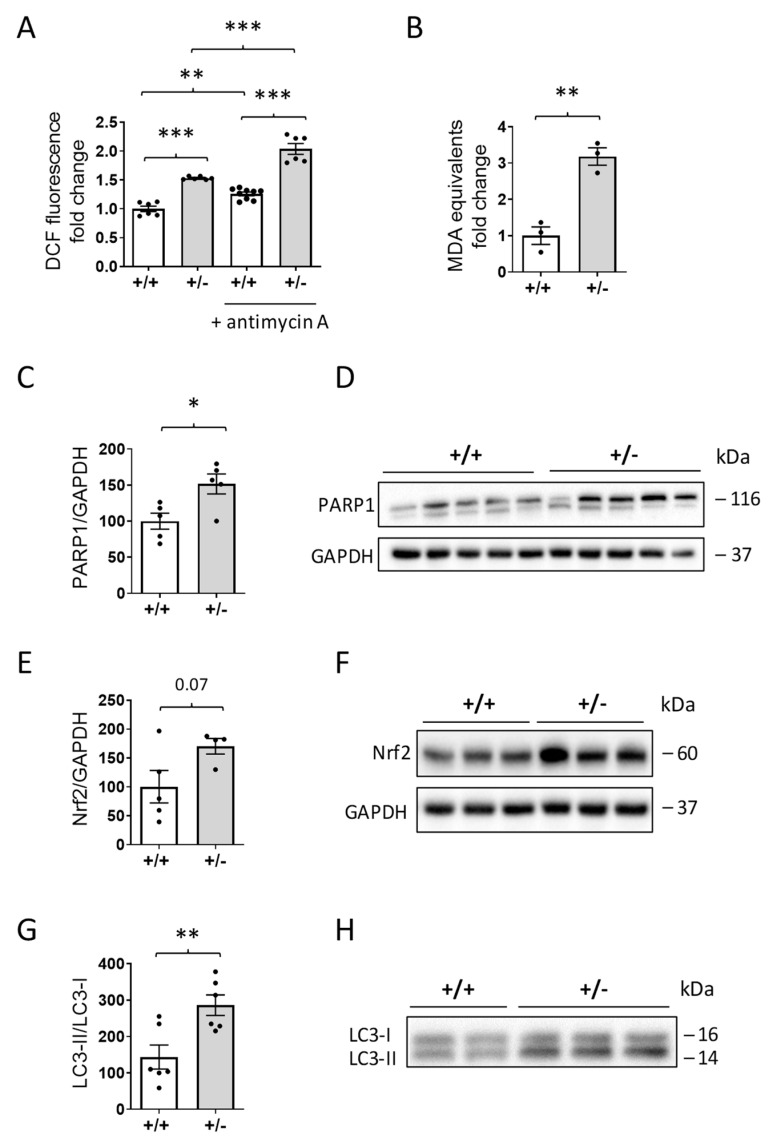
**Oxidative stress in the heart of *Cdkl5* +/− mice.** (**A**) Reactive oxygen species (ROS) production in isolated heart mitochondria from *Cdkl5* +/+ and *Cdkl5* +/− mice measured using the fluorescent probe DCFDA. The mitochondria were energized with succinate, and ROS production was determined by following the oxidation of H_2_DCF to its fluorescent form DCF. The specific inhibitor antimycin A was used to promote ROS production from mitochondrial complex III (n = 6). Data are expressed as fold change relative to controls and normalized to protein content; ** *p* < 0.01; *** *p* < 0.001 (Fisher’s LSD test after two-way ANOVA). (**B**) Measurement of the levels of lipid peroxidation biomarker malondialdehyde (MDA) in isolated heart mitochondria from *Cdkl5* +/+ and *Cdkl5* +/− mice (n = 3). Data are expressed as fold change relative to controls and normalized to protein content; ** *p* < 0.01 (two-tailed Student’s *t*-test). (**C**,**D**) Western blot analysis of poly(ADP-ribose) polymerase 1 (PARP1) levels in extracts of ventricular heart tissue from *Cdkl5* +/+ (n = 5) and *Cdkl5* +/− (n = 5) mice. The histogram in (**C**) shows PARP1 protein levels normalized to GAPDH. Examples of immunoblots for PARP1 and GAPDH of five animals from each experimental group are presented in (**D**). (**E**,**F**) Expression of nuclear factor erythroid 2-related factor 2 (Nrf2) in extracts of ventricular heart tissue from mice as in (**C**). The histogram in E shows quantification of Nrf2 protein levels normalized to GAPDH. Examples of immunoblots of three animals from each experimental group are presented in (**F**). (**G**,**H**) Expression of autophagy marker light chain 3 (LC3) in extracts of ventricular heart tissue from mice as in (**C**). The histogram in (**G**) shows the ratio of LC3II/LC3I. Examples of immunoblots for LC3 of three animals from each experimental group are shown in (**H**). Data in (**C**,**E**,**G**) are expressed as percentages of *Cdkl5* +/+ mice; values are presented as means ± SEM; * *p* < 0.05; ** *p* < 0.01 (two-tailed Student’s *t*-test).

**Table 1 ijms-24-05552-t001:** **Effect of genotype on ECG parameters.** Mean heart period duration (RR); mean duration of ventricular depolarization and repolarization (interval between Q and T waves) after applying Hodge’s formula to consider potential differences in RR values between groups (QTc); percentage of RR values that differ from the following values by more than 8 ms (pNN8); root mean square of the successive RR differences (RMSSD). All these values were computed during non-rapid-eye-movement sleep in *Cdkl5* +/+ (n = 8), *Cdkl5* +/− (n = 10) and *Cdkl5* −/− (n = 8) female mice. Values are represented as mean ± SE. * *p* < 0.05 vs. *Cdkl5* +/+, ** *p* < 0.01 vs. *Cdkl5* +/+, *** *p* < 0.001 vs. *Cdkl5* +/+ (Fisher’s LSD test after one-way ANOVA).

	*Cdkl5* +/+	*Cdkl5* +/−	*Cdkl5* −/−
RR interval (msec)	128.9 ± 2.6	115.7 ± 2.1 **	118.4 ± 3.1 *
QTc interval (sec)	0.749 ± 0.016	0.827 ± 0.015 **	0.820 ± 0.022 *
pNN8 (%)	30.3 ± 4.3	12.3 ± 2.7 ***	8.2 ± 2.8 ***
RMSSD (msec)	10.4 ± 1.3	5.8 ± 0.6 **	5.9 ± 0.9 **

**Table 2 ijms-24-05552-t002:** **Comparative expression of ion channel cardiac genes between *Cdkl5* +/+ and *Cdkl5* +/− mice.** mRNA expression levels of ion channel genes in the ventricle of 3-month-old *Cdkl5* +/+ and *Cdkl5* +/− mice. mRNA levels in tissues obtained from *Cdkl5* +/+ mice were set to 1, and relative expression levels in *Cdkl5* +/− mice are represented as mean ± SEM. Relative quantification was performed using the ΔΔCt method. The means of two stable reference genes (*Actin* and *Gapdh*) were used as normalization factors. * *p* < 0.05; n.s., not significant (two-tailed Student’s *t*-test).

Genes	*Cdkl5* +/+	*Cdkl5* +/−	*p*
*Kcnq1*	1.00 ± 0.119 (n = 7)	1.08 ± 0.048 (n = 9)	n.s.
*Kcnh2*	1.00 ± 0.055 (n = 7)	0.88 ± 0.032 (n = 9)	n.s.
*Kcnj2*	1.00 ± 0.069 (n = 7)	0.99 ± 0.037 (n = 9)	n.s.
*Scn5a*	1.00 ± 0.054 (n = 5)	0.87 ± 0.030 (n = 5)	*
*Hcn4*	1.00 ± 0.067 (n = 7)	0.76 ± 0.039 (n = 9)	*

**Table 3 ijms-24-05552-t003:** **Effect of genotype on heart weight and dimension.** Heart weight, body weight, heart over body weight ratio, and atrioventricular distance of 3–4-month-old *Cdkl5* +/+ (n = 18) and *Cdkl5* +/− (n = 27) mice. Values are represented as mean ± SEM; n.s., not significant (two-tailed Student’s *t*-test).

	*Cdkl5* +/+	*Cdkl5* +/−	*p*
Heart weight (mg)	110.83 ± 3.24	105.89 ± 2.27	n.s.
Body weight (g)	22.19 ± 0.27	21.74 ± 0.31	n.s.
Heart/body weight (mg/g)	4.98 ± 0.12	4.87 ± 0.07	n.s.
Atrioventricular distance (cm)	0.51 ± 0.01	0.52 ± 0.03	n.s.

## Data Availability

The datasets analyzed during the current study are available from the corresponding author upon reasonable request.

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
