# Peer review of "Cardiac Functional and Structural Abnormalities in a Mouse Model of CDKL5 Deficiency Disorder"

_ijms, 2023, doi:10.3390/ijms24065552_

Round 1
Reviewer 1 Report
Cdkl5 deficiency disorder (CDD) is a severe X-linked neurological condition affecting primarily young female patients. Recent evidence suggests CDD affects other tissues including the heart. Here, Loi and colleagues present the first characterization of the cardiac phenotype that manifests in female mice heterozygous for Cdkl5. A wide survey of cardiac phenotypes are assayed and it is found that Cdkl5+/- mice have abnormal indicators of heart rhythm, fibrosis, ion channel expression, reactive oxygen species, mitochondrial function, and autophagy. The paper is well written, and the authors should be commended for including all methodological details. The paper could benefit from additional data interpretation and targeted experiments that shed light into Cdkl5 mechanisms.
Major points:
1. Although many cardiac changes are surveyed and presented, it is not clear how the results are interpreted or what the primary versus secondary consequences of Cdkl5 deficiency are. Can the phenotype be rescued by targeting any of the observed changes?
2. Investigation of possible direct mechanisms regulating Cdkl5 and/or its downstream phosphorylation targets would be helpful. What is already known and about these mechanisms and how could they contribute to the observed cardiac phenotypes? For example, does the known phosphorylation of the epigenetic regulator MeCP2 by Cdkl5 have a role here? If so, RNA-seq analysis may be quite informative over sampling a few genes by qPCR and immunoblotting.
3. What are the cardiac phenotypes of Cdkl5-/- null females? Are phenotypes more severe than in heterozygous mice?
4. CDD also is reported for hemizygous males. It appears that male mice were not studied in this manuscript. However, it would be of interest to show whether the cardiac abnormalities observed in female hearts also occur in males.
5. Figure 1. What are the RR and QTc intervals for Cdkl5+/+ versus +/- hearts after atropine and atenolol treatment? By comparing baseline conditions (panels 1C-D) to the changes from saline (panels 1G-H), it looks like intervals after atropine and atenolol may be similar between the genotypes. If true, would this argue against a possible neuromodulatory contribution, and rather suggest Cdkl5 hearts are set to a parasympathetic-like state while still responding normally to atenolol?
6. It is hypothesized that Cdkl5 accelerates aging in hearts. Data across multiple ages are needed to support this idea beyond those presented from a single age group (3-4-month-old animals). The observations presented here are much more in line with a general decline in cardiac function that commonly occurs in multiple forms of heart failure. Do Cdkl5 animals succumb to heart failure?
7. Similarly, hypothesized senescence is difficult to demonstrate in the heart and is not supported without presenting supporting data. Adult cardiomyocytes do not undergo appreciable cell division, and the increase in fibrosis suggests fibroblasts are proliferating rather than senescing.
8. When stated, figure legends describe significance as measured by t-test, even those with 3 or more groups. ANOVA is more appropriate.
Minor points:
1. Figure 1B: The mean for Cdkl5+/+ animals (100%) in the Cdkl5 qPCR does not look to be the average of the n=6 data points (~75%?).
2. An antibody against Cdkl5 is listed in the reagent table but not used. Is Cdkl5 detectable in the heart? Does it localize to specific cell types and/or subcellular locations?
3. The Akt antibody is listed as #4061 from Cell Signaling, but #4061 is an antibody against N-Cadherin.
4. What is the rationale for blotting for Erk1/2 phosphorylation (Figure 4)?
5. Are the crystalline-like inclusions observed in mitochondria known to affect respiration and/or ROS production?
Reviewer 2 Report
Loi and cols. report in this manuscript the finding of cardiac alterations in a model of CDKL5 deficiency, broadening the effects of this factor to non-neuronal tissues. The study addresses several functional and structural analysis and in general is well performed and the results are convincing and well described.
Minor points:
1.- In figure 6 legend, when it says “Data in C, E, and H are expressed…” it should say “Data in C, E, and G are expressed…”
2.- In this same figure (Fig. 6), the results shown in panel H are not very convincing: only in one sample out of three from the +/- genotype (the one closer to the kDa position, right) shows a clear increase in the proportion of LC3-II vs LC3-I, represented quantitatively in panel G. The other two samples actually seem to go in the opposite direction. Please check. A positive control, where a clear increase in autophagy (and hence in LC3-II) is induced would also help to clarify these results.
3.- Nrf2 is usually related to a mitochondrial biogenesis increase. It seems somehow contradictory that its expression is increased in the +/- genotype while at the same time the mitochondrial respiratory activity is diminished. Please discuss this point.
4.- Some discussion speculating on the molecular mechanistic insight, trying to connect the observed alterations could be added (what could be the primary targets driving the different consequences observed, and what could be secondary effects).
Round 2
Reviewer 1 Report
The authors were responsive to most of my original concerns. The inclusion of additional ECG parameters is appreciated and helps to provide a clearer picture of the heterozygous phenotype relative to human pathology and that of homozygous animals. While the revisions significantly strengthen the paper, the lack of mechanistic insight could still be considered a weakness.